# Mechanistic insight on water dissociation on pristine low-index $TiO_2$ surfaces from machine learning molecular dynamics simulations

Zezhu Zeng [1], Felix Wodaczek [1], Keyang Liu[2], Frederick Stein [3,4], Jürg Hutter[3], Ji Chen [2,5,6] & Bingqing Cheng [1]✉

Water adsorption and dissociation processes on pristine low-index $TiO_2$ interfaces are important but poorly understood outside the well-studied anatase (101) and rutile (110). To understand these, we construct three sets of machine learning potentials that are simultaneously applicable to various $TiO_2$ surfaces, based on three density-functional-theory approximations. Here we show the water dissociation free energies on seven pristine $TiO_2$ surfaces, and predict that anatase (100), anatase (110), rutile (001), and rutile (011) favor water dissociation, anatase (101) and rutile (100) have mostly molecular adsorption, while the simulations of rutile (110) sensitively depend on the slab thickness and molecular adsorption is preferred with thick slabs. Moreover, using an automated algorithm, we reveal that these surfaces follow different types of atomistic mechanisms for proton transfer and water dissociation: one-step, two-step, or both. These mechanisms can be rationalized based on the arrangements of water molecules on the different surfaces. Our finding thus demonstrates that the different pristine $TiO_2$ surfaces react with water in distinct ways, and cannot be represented using just the low-energy anatase (101) and rutile (110) surfaces.

Titanium dioxide ($TiO_2$) interfaces with water have paramount technological importance in photocatalysis, catalyst support and medical applications[1–3], and also serve as a prototype system in surface science[4]. However, even for defect-free stoichiometric interfaces, water dissociation is far from being well-understood, let alone surfaces with defects[5,6], polaronic effect[7], or reconstructions[8].

Past studies exclusively focus on anatase (101) and rutile (110), as they have the lowest surface energy for each phase and are thus most abundant in nature[2]. There remain many controversies. For rutile (110),

scanning tunneling microscopy (STM) studies indicated that water dissociation happens at defect sites[9], while x-ray photoelectron spectroscopy[10] found water dissociation on the hydrated stoichiometric surface at various coverages and temperatures. Experiments using both supersonic molecular beam and STM revealed a dynamic equilibrium of water dissociation at low temperature and water coverage[11], although oxygen vacancy is inevitable on the sample surface. From the theory side, static density functional theory (DFT) calculations[12,13], molecular dynamics (MD) simulations based on DFT[12]

[1]The Institute of Science and Technology Austria, Am Campus 1, 3400 Klosterneuburg, Austria. [2]School of Physics, Peking University, Beijing 100871, P. R. China. [3]Department of Chemistry, University of Zurich, Winterthurerstrasse 190, 8057 Zurich, Switzerland. [4]Center for Advanced Systems Understanding (CASUS), Helmholtz-Zentrum Dresden, Rossendorf (HZDR), Untermarkt 20, 02826 Görlitz, Germany. [5]Interdisciplinary Institute of Light-Element Quantum Materials and Research Center for Light-Element Advanced Materials, Peking University, Beijing, China. [6]Frontiers Science Center for Nano-Optoelectronics, Peking University, Beijing, China. ✉e-mail: bingqing.cheng@ist.ac.at

or machine learning potentials (MLPs)[14,15] have debated severely about the exact fraction of water dissociation on rutile (110) surface, and the results are sensitive to the underlying functionals and simulation setups. For instance, very recently, MD using PBE-D3 MLP predicted a fraction of only 2%[14], while SCAN MLP obtained 22%[15]. For anatase (101), previous STM experiment suggested that water adsorbs molecularly on almost defect-free surface[16] or reduced surface with subsurface defects[17] in ultrahigh vacuum, but synchrotron radiation photoelectron spectroscopy[18] observed a water monolayer on the stoichiometric surface involves both molecular and dissociative adsorption, and X-ray diffraction experiments[19] also showed water dissociation on reduced surfaces with both ultrathin and bulk water. In simulations, BLYP MD[20] and DFT MD[21] with optB86b-vdW functional both predicted that bulk water adsorbs molecularly on (101) surface, while static DFT calculations with PBE functional showed the coexistence of dissociated and molecular water at monolayer coverage[22]. Recently, Andrade et al.[23] used a combination of MLPs with SCAN functional and enhanced sampling MD, and predicted a water dissociation fraction of 5.6%. Li et al.[24] reported a slightly higher fraction of 7.8% using DFT MD with PBE functional.

For other high-energy surfaces, studies are relatively rare and even less is clear regarding water dissociation[2], although these surfaces are crucial to investigate as they may have higher catalytic activity than the stable surfaces[25]. As with the stable surfaces, different functionals provide different pictures for surface energy and water absorption, for example, for the rutile (100) surface[26–28]. In addition, surfaces with high reactivity such as anatase (110) and rutile (001) decrease rapidly during the crystal growth process[2], and surfaces including anatase (001)[29], and rutile (011)[30] can have spontaneous reconstructions in vacuum, both of which greatly hinder the preparation of the pristine surfaces. On the other hand, a reconstructed surface can be lifted to the unreconstructed state at aqueous environments, for example for rutile (011)[31], establishing the importance of studying the high-energy pristine surfaces.

Understanding water interactions with pristine TiO$_2$ interfaces is difficult: In experiments, preparing pristine surfaces[32], preventing contamination in dipping experiments[33], and step-by-step characterizing water adsorption[34] on pristine surfaces in aqueous environments are all challenging. High-energy surfaces are even more difficult to investigate experimentally due to the high surface activity and low stability[3]. In simulations, empirical forcefields lack qualitative accuracy and do not allow water dissociation[35]. DFT calculations are restricted in system size, time scale, and the approximation of the exchange-correlation functional. Machine learning potentials[36,37] allow converged MD simulations with ab initio accuracy, but previous MLPs work only for either anatase (101)[23] or rutile (110)[14,15,38]. Thus a complete description for the interactions between water and various low-index TiO$_2$ is still missing, along with a mechanistic understanding of water dissociation.

Herein, we constructed MLPs that can simultaneously describe bulk anatase, rutile, bulk water, and bulk water-TiO$_2$ and vacuum-TiO$_2$ interfaces for anatase (001), (100), (101), (110) and rutile (001), (100), (011), (110) surfaces. We considered three different DFT functionals, and exploited committee models[39] to provide error estimates of the MLPs. We then computed the free energies of water dissociation at various interfaces, providing a quantitative estimate of how much water dissociation occurs. Finally, we developed a machine-learning-based algorithm to identify the dissociation mechanism and proton transfer pathways automatically, and rationalized the different mechanisms based on the atomic structures of the interfaces.

## Results
### Water adsorption and dissociation
We systematically investigate the influence of the underlying DFT functionals on water interactions with the TiO$_2$ surfaces. We first fitted

a committee model[38] made of four fits of the MLP trained on the optB88-vdW DFT functional. We then fitted a set of Δ-learning committee MLPs based on the difference between the SCAN and the optB88-vdW potential energy surfaces, and another set based on the difference between the PBE and the optB88-vdW potential energy surfaces for the bulk TiO$_2$-water interface systems. One can then use these Δ-learning potentials on top of the optB88-vdW baseline to obtain the atomic interactions at the PBE or SCAN level of theory. In MD simulations, we employed the three sets of committee MLPs based on the SCAN, PBE, and optB88-vdW functionals.

To reversibly sample water dissociation, we employed well-tempered metadynamics[40] simulations with adaptive bias[41]. The collective variable (CV) is the minimal distance $S_{O\text{-}H}$ of a surface oxygen to any hydrogen in the system, which is the same as ref. 23. During a metadynamics run, a time-dependent bias potential $V(S_{O\text{-}H}(t))$ is added to the Hamiltonian of the system $\mathcal{H}$, i.e. $\mathcal{H}_{biased} = \mathcal{H} + V(S_{O-H}(t))$. This bias distorts the equilibrium probability distribution, and the unbiased ensemble averages for an observable $O$ can be obtained with a reweighting procedure[42]:

$$<O> = \frac{<Oe^{\beta V(t)}>_{biased}}{<e^{\beta V(t)}>_{biased}},\qquad(1)$$

where $< \cdot >$ indicates the ensemble average sampled using the corresponding Hamiltonian. The free energy surfaces (FES) with respect to the CV can thus be calculated from the unbiased histogram of $S$.

In the metadynamics simulations, each system contains 128 water molecules and about 200 TiO$_2$ atoms. The bulk water in the center of the simulation box has a density $1\,\mathrm{g\,mL^{-1}}$. In simulations using the optB88-vdW or the SCAN MLPs, the temperature was kept at 330 K, which is 30 K higher than room temperature in order to roughly account for the nuclear quantum effects in room-temperature water as used in ref. 23. For the PBE MLPs, the simulation temperature was elevated to 370 K to avoid water freezing, as PBE water has a high melting point of about 417 K[43]. We performed simulations on pristine anatase (001), (100), (101), (110) surfaces and rutile (001), (011), (100), (110) surfaces. The metadynamics simulations for the anatase (001) surface show a lot of hysteresis so the computed FES lacks convergence, probably due to that the CV neglects certain degrees of freedom relevant to water dissociation on this surface, so we removed it from further analysis.

For rutile (110), previous calculations predicted that water interaction with the slab has an odd-even oscillation behavior with respect to the number of O-Ti-O trilayers[12,14,15]. As detailed in the Supplementary Information, we observed the same oscillation in the MLP MD simulations, and thus used 10 trilayers in the productions runs to ensure the convergence with respect to the slab thickness. For the anatase (100), (101), (110) surfaces and rutile (001), (011), (100), no evident dependence of water dissociation on slab thickness was observed in our simulations using different number of layers (Supplementary Information).

Snapshots of atomic configurations for the seven TiO$_2$-water interfaces from the optB88-vdW MLP metadynamics simulations are shown in Fig. 1a. On anatase (110) and rutile (001), two water molecules are adsorbed simultaneously by each surface undercoordinated four-fold (Ti$_{4c}$) site. For the other five surfaces, one water molecule is adsorbed on each five-fold (Ti$_{5c}$) site. The O atoms in these adsorbed water molecules occupy the missing oxygen sites of TiO$_2$ while the H atoms point away from the surface. We thus classify adsorbed water molecules (H$_2$O-Ti) when the Ti-O distance is within 2.65 Å. We also define the first-layer water (H$_2$O$^{(1)}$) as non-adsorbed water molecules close to the surface, here classified based on within 3.5 Å of the undercoordinated two-fold O$_{2c}$ sites. Surface O$_{2c}$ atoms can accept protons to form the bridging hydroxyl groups (H-O$_r$), and terminal hydroxyl groups on surface Ti atoms can emerge (HO-Ti). For liquid

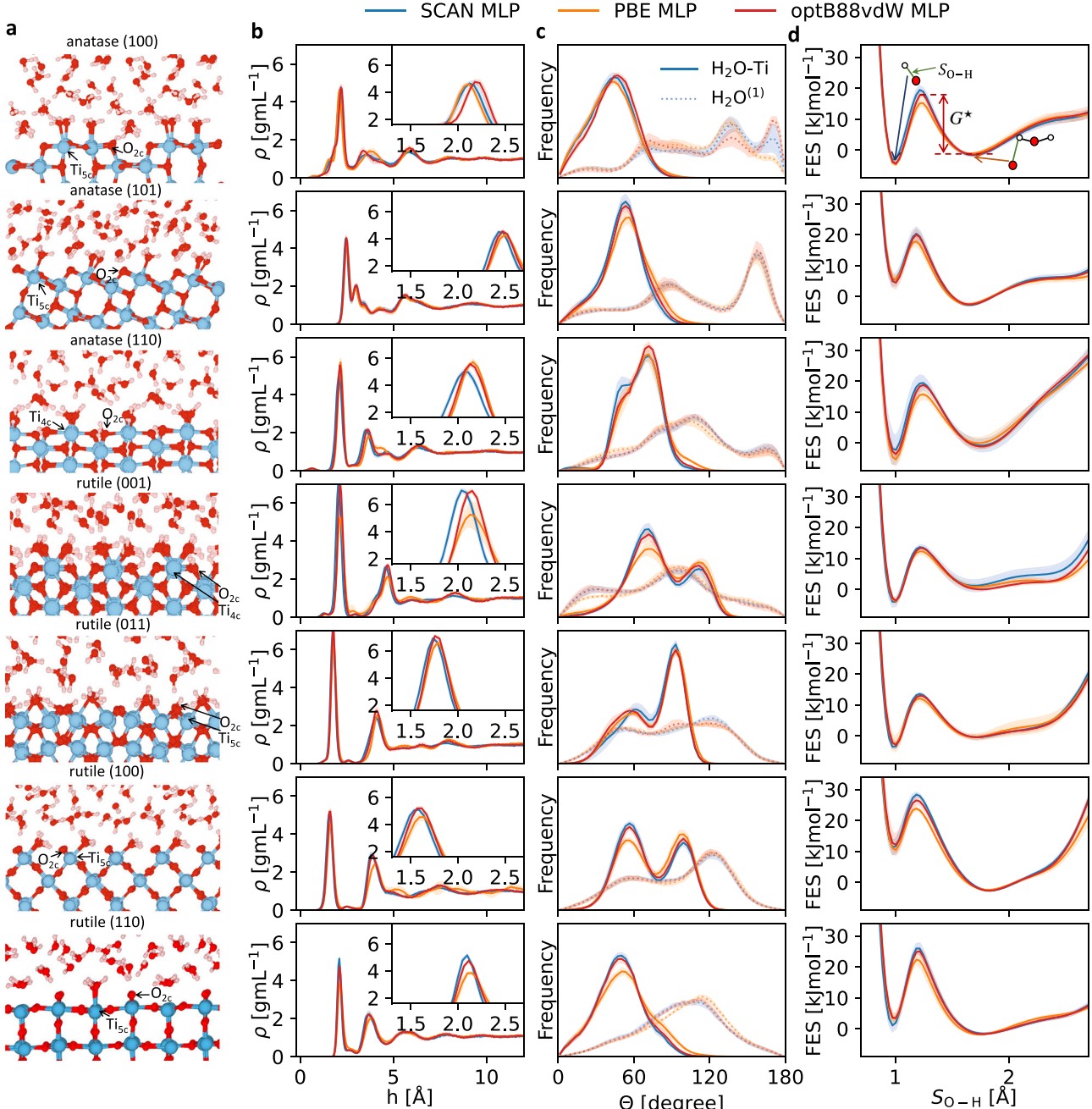

**Fig. 1 | Adsorption and dissociation of water on pristine low-index TiO₂ surfaces. a** Snapshots of atomic positions for anatase (100), (101), (110) and rutile (001), (011), (100), (110) surfaces in water. Surface undercoordinated four-fold $Ti_{4c}$, undercoordinated five-fold $Ti_{5c}$, undercoordinated two-fold $O_{2c}$ (also known as oxygen bridge site), and coordinated three-fold $O_{3c}$ sites are indicated. **b** The water density profiles $\rho$ as a function of the vertical height $h$ from the outmost Ti layer on surfaces. **c** The orientation distributions of water molecules near the surface, for the water adsorbed on surface Ti (solid curves) and first-layer water (dashed curves). $\theta$ is the angles between the water dipole vector and the surface norm. **d** The free energy surface (FES) as a function of the minimal distance $S_{O-H}$ (marked as green solid lines) of a surface $O_{2c}$ atom to any hydrogen in the system. The two valleys on the FES correspond to molecular and dissociated water states as schematically indicated. In (**b**–**d**), results are from three committee machine learning potentials (MLPs) based on SCAN, PBE and optB88-vdW density functionals. Each committee MLP has 4 individual MLPs, and the thick lines show the average estimate from the four, while the shaded areas show their standard deviations.

water farther from the surface, no isolated OH or H₃O groups are observed.

Water adsorption on these surfaces is characterized by the density profiles $\rho$, as shown in Fig. 1b. Comparing the density profiles computed using the MLPs based on the three DFT functionals, the differences are relatively small between SCAN MLPs and optB88-vdW MLPs, while PBE MLPs consistently predict weaker water adsorption suggested by the lower height of the first peak. Each density profile exhibits a prominent first peak near the surface, and lower subsequent peaks. This suggests a highly structured arrangement in the water close to the surfaces, with decaying order going into the bulk. Such interface-induced structuring can affect the water up to about 10 Å away from the surface. The first water density peak is more pronounced on the four rutile surfaces than for the three anatase surfaces. For anatase (100), (110), and rutile (001), the tiny bumps in the density profiles close to the surfaces are due to hydroxyls formed on $O_{2c}$ sites.

The atomic configurations in Fig. 1a help to rationalize water structuring near the interfaces. For most surfaces, the first and the

second peaks in the density profiles (Fig. 1b) correspond to the adsorbed water ($H_2O$-Ti) and first-layer water ($H_2O^{(1)}$), respectively. However, for the anatase (100), both the $H_2O$-Ti and $H_2O^{(1)}$ contribute to the first peak, due to the relatively large gaps between the surface $Ti_{5c}$ sites, which provides adequate spaces for $H_2O^{(1)}$ to be closely attracted to $O_{2c}$ sites. The same reason explains the proximity between the first two density peaks in anatase (101). For rutile (001), the second density peak is particularly far from the surface (-5 Å). This is because two water molecules with different orientations can be simultaneously adsorbed onto the same surface $Ti_{4c}$ atom (as shown in Fig. 1a). These water molecules form a close and dense $H_2O$-Ti layer and hinder the surface attraction for the $H_2O^{(1)}$ layer.

We further characterize the structure of interfacial water via their orientations, defined as the dipole directions - the angles ($\theta$) between the dipole vector (oxygen pointing to the mid-point of two hydrogens) and the surface norm. Figure 1c shows the orientation distribution for $H_2O$-Ti and $H_2O^{(1)}$ separately: The solid curves are for $H_2O$-Ti, and dashed curves are for $H_2O^{(1)}$. For $H_2O$-Ti, as hydrogen atoms point away from the nearest Ti atoms, the distributions of $\theta$ are dominated by acute angles. $\theta$ for rutile (001), (011) and (100) have double peaks, as the dipole vectors of $H_2O$-Ti can point along both sides of the surface. This double-peak feature was also reported in a MD study using an empirical forcefield from Kavathekar et al.[44], suggesting that it is probably insensitive on the underlying potential surfaces assumed. For $H_2O^{(1)}$, the dipole vectors usually point downwards, inducing an obtuse-angle-dominated distribution for the $\theta$. As we will later show, such downwards orientations may be relevant for proton transfers.

The equilibrium ratio between surface hydroxyl and molecular water at an $O_{2c}$ site can be determined as $f = \exp(-\beta\Delta G)$, where $\Delta G$ is their free energy difference. This is revealed by the free energy surface as a function of the CV (shown in Fig. 1d). $S_{O \cdot H} \approx 1$ Å means a surface oxygen has formed a hydroxyl group with a hydrogen atom from water, and $S_{O \cdot H} \approx 1.75$ Å means the closest water remains molecular. All three sets of MLPs based on the different functionals give quite consistent results for the FES of water dissociation on seven surfaces, while different surfaces have distinct FES for water dissociation and adsorption. Comparing $\Delta G$ for anatase $TiO_2$ facets at $S_{O \cdot H} = 1$ Å and 1.75 Å, we conclude that on (100) and (110) dissociative adsorption is preferred, while on (101) molecular adsorption is more common. Our conclusion for anatase (101) is consistent with previous calculations[23,24]. For rutile, on (001) and (011) dissociation is favorable, and on (100) molecular adsorption is highly preferred. For rutile (110), with the thick slab of 10 trilayers, the $\Delta G$ between dissociated water and molecular water is $5.2 \pm 0.6$ kJmol$^{-1}$ (with molecular state being more stable) at the optb88-vdw MLP level, $2.6 \pm 1.2$ kJmol$^{-1}$ at the SCAN MLP level, and $5.1 \pm 0.6$ kJmol$^{-1}$ at the PBE MLP level. Our results thus agree with previous simulations that rutile (110) favors molecular adsorption[12,14,15], and are also consistent with STM[11] and x-ray photoelectron spectroscopy[10] experiments which suggest that the energy difference between the molecular and the dissociated state is small.

Figure 1d also shows the free energy activation barrier ($G^\star$) for molecular water to dissociate. For example, for anatase (101) (see Table S3 for the $G^\star$ of other surfaces), the $G^\star$ is $23 \pm 2$ kJmol$^{-1}$, in good agreement with the value from $G(S_{O \cdot H}(t))$ in ref. 23. This $G^\star$ is about 10 times the thermal energy at room temperature. AIMD simulations are restricted to the picosecond timescale, which is probably inadequate to overcome the large $G^\star$ and obtain reliable statistics regarding water dissociation. In contrast, Our metadynamics simulations can freely diffuse across the barrier and reliably estimate the FES.

## Pathway for proton transfer and water dissociation
The atomic pathway of proton transfer is important for understanding water dissociation on $TiO_2$, but the analysis is nontrivial and generally needs a case-by-case consideration exploiting physical and chemical insights. For anatase (101), Andrade et al.[23] provided a detailed proton

transfer mechanism, using hand-crafted CVs inspired by earlier computer simulations of proton diffusion in aqueous solutions[45]. For other surfaces, however, the proton transfer pathway is largely unknown, and it is unclear whether a fixed set of CVs is sufficient to capture all the possible mechanisms.

To investigate water dissociation mechanism in a general and automated way, we develop a machine-learning-based method. We take the last part of the trajectories from the optB88-vdW MLP metadynamics simulations with slow bias depositions, each contains 10,000 snapshots with a time step of 0.1 ps. The analysis focuses on the different atomic environments of hydrogen atoms in the system. Specifically, for a H atom in a certain frame of the metadynamics trajectory, we first compute a list of features $\chi$, including the H to its closest Ti distance (H-Ti), H to its closest neighboring H (H-H) and the second closest H distance, H to its closest O in $TiO_2$ (H-$O_t$) and its closest O in water, the surface normal of the displacement between H and its closest O, three proton transfer coordinates determined by the positions of the hydrogen, a donor oxygen atom O and an acceptor O′ (i.e. $\nu = d(OH) - d(O'H)$, $\mu = d(OH) + d(O'H)$, $r_{OO} = d(OO')$)[46]. We then use sparsified kernel Principal Component Analysis (kPCA) based on these features $\chi$: we build support vectors by selecting a small set of H environments using farthest point sampling, build the kPCA map using cosine kernel, and finally project the $\chi$ of all the H environments onto the saved support vectors. The kPCA maps visualize the similarity between different hydrogen atomic environments, and the axes of the maps capture the most important variance within the data points[47]. Such procedures allow us to compare the H environments of different systems with various $TiO_2$ surfaces on the same footing. The whole procedure is streamlined by the ASAP package[48].

In Fig. 2 we show the kPCA plots of the hydrogen atomic environments in water-rutile (110) system, and the plots for other facets can be found in the Supplementary Information. Each dot on the plot indicates the environment of each hydrogen atom. The kPCA plots can be rationalized using different color coding based on the various features. Four selected panels are shown in the Fig. 2, and the rest of the kPCA plots are provided in the Supplementary Information. The whole set of H environments forms well-separated clusters, and each cluster corresponds to a H in a specific state (see Fig. 2a): e.g. H adsorbed on the surface O (H-$O_t$), OH adsorbed on Ti (HO-Ti), adsorbed $H_2O$ ($H_2O$-Ti), first-layer $H_2O$ ($H_2O^{(1)}$), and $H_2O$ farther from the surface ($H_2O^{(>1)}$). These different states are illustrated in Fig. 2e, and the classification scheme is described in the Supplementary Information. Whether a hydrogen atom is in a hydroxyl rather than a water molecule is suggested by a large value of the H-H distance greater than about 1.8 Å (Fig. 2b). The hydrogen-bonded complexes appear at the indicated places on the edge of the clusters. Within each cluster, the variability mainly comes from the orientation of the water molecule. For example, the H atoms in $H_2O$-Ti can point towards or away from surface oxygen atoms (see Fig. 1a), causing the gradients in the H to $O_t$ distance (see Fig. 2d). $H_2O^{(1)}$ can have hydrogen up or down (see Fig. 1a), which explains the variance (see Fig. 2c) in the H-Ti distances within the corresponding cluster.

From the kPCA coordinates and the weighted frequency count, one can build FES for these generalized coordinates using Eqn. (1), as shown in Fig. 2e, which demonstrates the relative probability of the H in different states. Note that, for the free energy difference between surface OH and $H_2O$, this FES is different from the one in Fig. 1d as the former also considers the configurational entropy coming from the number of possible sites.

We then consider the time dependence of the H environments, in order to reveal hydrogen transition pathways during the MD simulations. In Fig. 3a–b, two representative systems, rutile (001) and anatase (101), are used to show two different transition pathways between different states (illustrated in Fig. 2e). If a hydrogen atom transits between different states, a gray line is drawn between the initial and

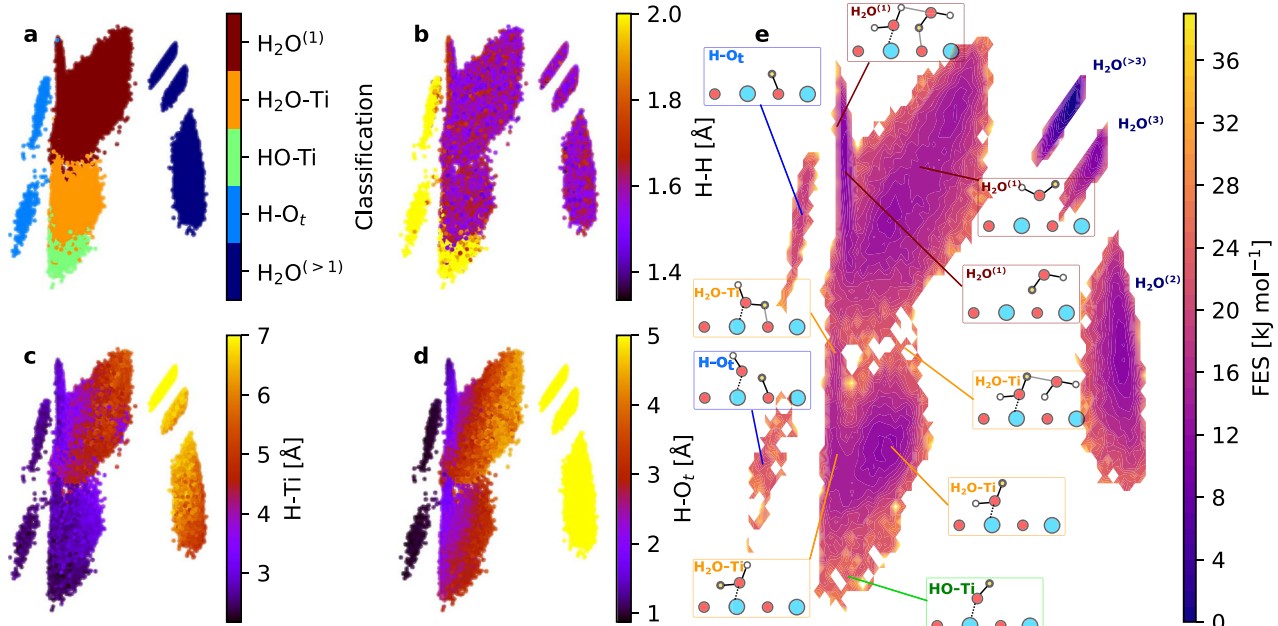

**Fig. 2 | Analysis and visualization for the hydrogen environments in water-rutile (110). a–d** kernel Principal Component Analysis (kPCA) maps of all the atomic environments of hydrogens in the system, colored according to different attributes: classification of H environments (**a**), the distance of H to its closest neighboring H (H-H) (**b**), the distance of H to its closest Ti (H-Ti) (**c**), the distance of H to its closest O in TiO$_2$ (H-O$_t$) (**d**). **e** The free energy surface (FES) as a function of the two principal axes of the kPCA map of the hydrogen environments. Representative atomic configurations are illustrated in the insets, with the hydrogen atom associated with the indicated H environment highlighted in orange circles. H atoms in the water layers farther away from the surface are annotated as H$_2$O$^{(2)}$, H$_2$O$^{(3)}$ and H$_2$O$^{(>3)}$. The dashed bonds indicate a water molecule or hydroxyl (OH) is adsorbed onto a surface Ti, and the gray bonds denote hydrogen bonds.

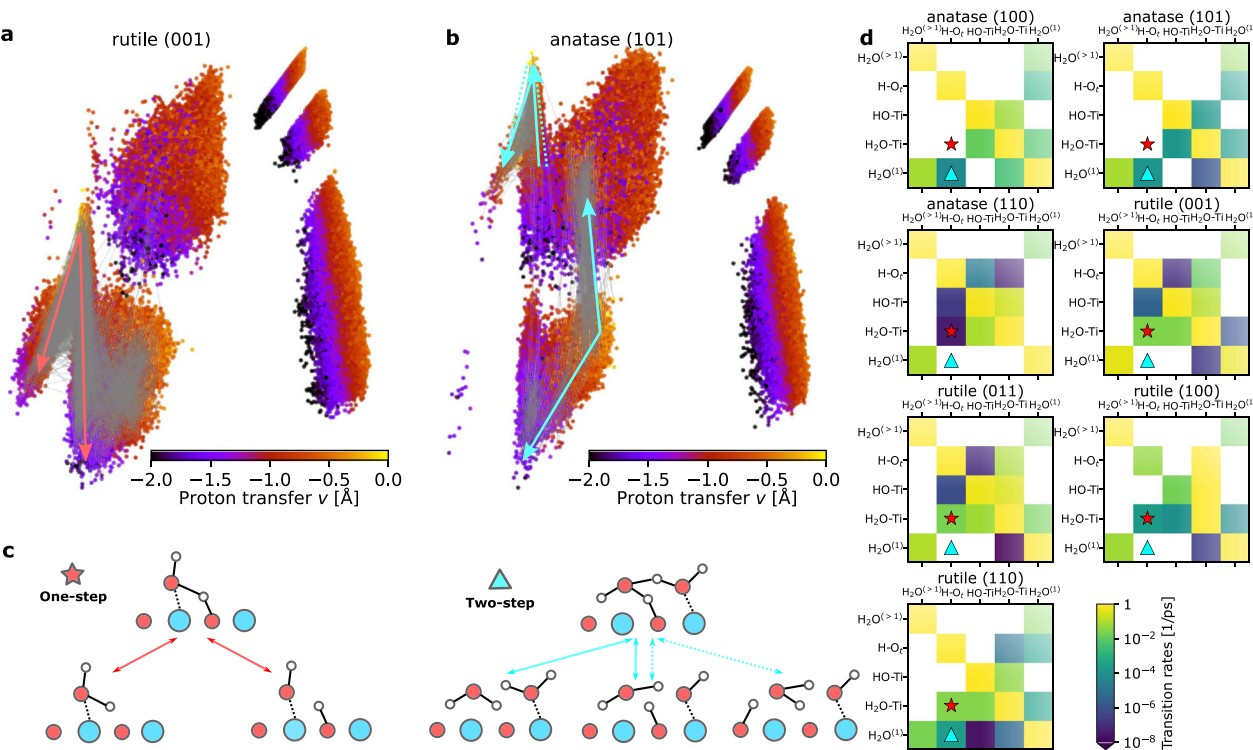

**Fig. 3 | Proton transfer mechanism in water dissociation on pristine low-index TiO$_2$ surfaces. a–b** kernel Principal Component Analysis (kPCA) plots for hydrogen environments in water-rutile (001) (**a**) and water-anatase (101) (**b**), colored according to the proton transfer coordinates $v$. A high $v$ indicates that a proton is in the middle of being transferred. The one-step and the two-step mechanisms are indicated using the red and the cyan arrows, respectively. **c** A schematic of one-step water dissociation (red solid arrows), two-step water dissociation (cyan solid arrows) and proton transfer (cyan dashed arrows) mechanisms. **d** H transition probability between different states, computed from 10,000 metadynamics snapshots that are 0.1 ps apart. The red stars indicate the matrix elements that are signature of the one-step transition mechanisms, H from H$_2$O-Ti becomes O$_t$; The cyan triangles indicate the signature element for the two-step process, H from H$_2$O becomes H$_2$O$^{(1)}$. Only elements in the lower triangle of the matrix are marked.

the final environments. For clarity, we only plot the transition lines for hydrogens near the surface, i.e. in H·$O_t$, HO-Ti, $H_2O$-Ti, or $H_2O^{(1)}$ states, because farther water molecules ($H_2O^{(>1)}$) do not participate in the water dissociation reactions. The two surfaces reveal two distinct modes for water dissociation and proton transfer. Rutile (001) has an one-step water dissociation process illustrated by the solid red arrows Fig. 3c: a water molecule adsorbed in surface Ti directly splits into a surface OH on Ti and a H on $O_{2c}$. The transition state is a $H_2O$-Ti that is hydrogen-bonded to a surface $O_{2c}$. For anatase (101), the gray transition lines are consistent with a two-step proton transfer process, marked using thick cyan arrows. This two-step process is schematically illustrated using the solid cyan arrows in Fig. 3c: a water molecule adsorbed in surface Ti ($H_2O$-Ti) donates a H to a first-layer water molecule ($H_2O^{(1)}$), and the latter transfers another H to a surface $O_{2c}$ site to form a surface hydroxyl. The transition state has an intermediate water molecule that forms hydrogen bonds to both $O_{2c}$ and $H_2O$-Ti. The analogous mechanism also serves for proton transport between different $O_{2c}$ sites, indicated using the dashed blue arrows in Fig. 3c. In the Supplementary Information, we supply an algorithm that can further distinguish between the recombination/dissociation events and pure proton transport that does not change the amount of surface hydroxyl coverage. Andrade et al.[23] reported the same two-step mechanism for anatase (101) in MLP MD simulations. For both the one-step and two-step mechanisms, the transitions of hydrogen happen via proton transfer, as revealed by the proton transfer coordinate $\nu$ used as the color scale in Fig. 3a–b. The critical difference between the two mechanisms is the participation of $H_2O^{(1)}$. Another distinction is that, after an one-step dissociation event the H·$O_t$ and HO-Ti are next to each other, while after a two-step dissociation the separation can be larger.

To quantify proton transition rates, we assume quasi-equilibrium in the dynamics[49] of the metadynamics simulations with slow bias depositions. The unbiased transition probability from one state $x_a$ to another state $x_b$ after time $\Delta t$ is

$$K(x_b, \Delta t | x_a, 0) = \frac{<\delta(x(t+\Delta t) - x_b)\delta(x(t) - x_a)e^{\beta V(t+\Delta t)}>}{<\delta(x(t) - x_a)e^{\beta V(t)}>}, \quad (2)$$

where the Dirac delta functions $\delta(x(t) - x_a)$ and $\delta(x(t+\Delta t) - x_b)$ select the segment of trajectories that are starts from $x_a$ in time $t$ and ends at $x_b$ in time $t + \Delta t$, respectively. Here the $\Delta t = 0.1$ ps is the time between two subsequent MD snapshots.

Figure 3d shows the transition matrices between different states of hydrogen atoms for the seven $TiO_2$ surfaces. The color scale of the matrix element in row $x_a$ and column $x_b$ indicates the probability of a hydrogen atom that is in state $x_a$ at $t$ going to state $x_b$ at $t + \Delta t$. Most hydrogen atoms remain in their original state within the $\Delta t = 0.1$ ps, so the diagonal element of the transition matrix is typically close to one. The off-diagonal matrix elements correspond to hydrogen transits, from which one can infer the underlying mechanisms. To show this clearly, the red stars and cyan triangles in Fig. 3d indicate the signature matrix elements for the one-step and two-step transition mechanisms, respectively. These markers indicate whether the H atoms in H·$O_t$ are proton-transferred from $H_2O$-Ti or $H_2O^{(1)}$.

As we discuss below, for all the $TiO_2$ surfaces, different proton transfer and water dissociation mechanisms are related to the atomic arrangement: The one-step mechanism requires a close distance between surface $O_{2c}$ and the protons in water absorbed by surface Ti ($H_2O$-Ti), and the two-step mechanism needs the proximity of first-layer water ($H_2O^{(1)}$) to the surface.

Anatase (100) and (101) have only two-step transition process. The lack of the one-step process on these two anatase surfaces may be due to that the hydrogen atoms in $H_2O$-Ti point upwards (Fig. 1a), so $\theta$ in Fig. 1c adopt mostly acute angles and the distances between these H atoms and $O_{2c}$ are relatively large. Meanwhile, the $H_2O^{(1)}$ (Fig. 1a) are close to the surface with many H atoms pointing downwards,

facilitating the two-step proton transition mechanism. Anatase (110) is observed to have infrequent one-step process, as this surface is already densely covered by dissociated water in the metadynamics simulations, as also revealed from the FES in Fig. 1d.

Rutile (001), (011) and (100) surfaces exhibit relatively high rates, and the mechanisms are exclusively one-step. On these three facets, surface Ti atoms have strong adsorption of water, and many H atoms in $H_2O$-Ti point sideways as suggested by the double-peak feature of $\theta$ in Fig. 1c, which facilitates the H-bond formation and proton transfer with $O_{2c}$ sites. Meanwhile, $H_2O^{(1)}$ are relatively far from the surface, making clear gaps between the first and the second peaks in the density profiles (Fig. 1b) as previously discussed. The dense $H_2O$-Ti layer and the far $H_2O^{(1)}$ layer make the one-step process favorable and the two-step process unlikely. Rutile (110) has a coexistence of one-step and two-step processes, which may be explained by the intermediate Ti adsorption strength and $H_2O^{(1)}$ distances. The coexistence of both mechanisms was also observed in a recent MLP MD study on rutile (110) by Wen et al.[15]. Overall, rutile (001), (011) and (100) exhibits faster proton transit rates. The rate is strongly related to the free energy barrier from molecular water to surface hydroxyl as shown in Fig. 1d. Rutile (001) and (011) surfaces both own a relatively low $G^{\star}$ of about 13 kJmol$^{-1}$, which implies that water dissociation may happen faster. Rutile (011) has a unique corrugated surface structure with humps consisting of proton-accepting $O_{2c}$ sites (see Fig. 1a), which may help promoting water dissociation.

In summary, we constructed the first MLP that can simultaneously describe the interfaces between water and various anatase and rutile $TiO_2$ facets, pushing the limit of the capability of machine learning potentials for complex chemical systems.

Water dissociation fraction, free energy barrier and proton transfer on surfaces are key features for investigating the reactivity of $TiO_2$-water interfaces in chemical or photochemical settings, which is relevant for numerous practical applications[1,2]. Based on enhanced sampling MD simulations using the MLPs trained on three different DFT functionals, we resolved the long-standing debate about the state of water at different pristine $TiO_2$ surfaces: dissociative or molecular. In contrast to previous studies which almost exclusively focus on the anatase (101) and rutile (110) surfaces, we comprehensively elucidate water adsorption and dissociation processes on seven low-index surfaces in aqueous environments for the first time. We show that different pristine $TiO_2$ surfaces react with water in distinct ways, and cannot be represented using just the low-energy anatase (101) and rutile (110) surfaces. Surfaces such as anatase (100), (110) and rutile (001), (011) may be more reactive in photochemical water splitting than the stable surfaces as they favor more water dissociations. Our results thus imply that, in order to better understand the photocatalysis, catalysis and biomedical applications of $TiO_2$ (nano)particles, the high-energy surfaces need to be taken into account.

We further used a general and automated way to visualize and understand water dissociation and proton transfer mechanisms, based on the chemical features of protons. We rationalized the mechanisms based on the water arrangements on different surfaces. This not only allows a microscopic understanding of water interaction with these pristine interfaces, but also paves the way towards more complex surfaces with defects, polarons and reconstructions. The workflow can also be applied to other complex aqueous systems. For example, most solid surfaces under ambient conditions are covered by a thin film of water[50]. Other technologically relevant systems include: corrosion of steels, electrolysis of water on metal plates, confinement of water in two-dimensional materials[51].

## Methods
### DFT calculations
We used the CP2K package[52] for both DFT MD and single-point DFT calculations. The typical system size contains 64 water molecules and

about 200 $TiO_2$ atoms. For the optB88-vdW functional, we used a planewave energy cutoff of 350 Rydberg. We also tested a higher cutoff of 600 Rydberg: the difference in relative total energy is 0.25 meV/atom and the difference in force components is 20 meVÅ$^{-1}$ for configurations with about 300–400 atoms. Such differences are much smaller than the typical MLP training errors. For the single-point calculations using the SCAN functionals, we used a planewave cutoff of 1200 Rydberg, and for the PBE functional we used 600 Rydberg. The CP2K input files are provided in the SI repository.

## MLP

We generated flexible and dissociable MLPs based on optB88-vdW for the $TiO_2$/water system. The total number of configurations for the training set is 18930. We include pure water, and various flat and defective interfaces for anatase/rutile in vacuum and in bulk water. To effectively include configurations along transition paths of water dissociation and proton transfer, we performed 4 iterative rounds in constructing the MLPs: configurations were selected from the metadynamics simulation trajectories generated by a previous generation of the MLP, and then recomputed using optB88-vdW DFT and added to be training set. The training errors for energy and atomic force components are 1.5 meV/atom and 133 meVÅ$^{-1}$, respectively. The testing errors for energy and atomic force components are 1.6 meV/atom and 130 meVÅ$^{-1}$, respectively. This set of MLPs work for: (i) Bulk water and water/vapor interface; (ii) Pristine anatase (101), (001), (110), (100) and rutile (011), (110), (001), (100) surfaces, in vacuum and in bulk water; (iii) These eight surfaces with some simple stoichiometric surface defects, in vacuum and in bulk water.

The surface defects are restricted to the type by removing a multiple of $TiO_2$ formula units, so no polaronic effects that stem from oxygen vacancies are considered. Although the present study focuses on pristine surfaces, the benchmarks for the MLPs on defected surfaces in bulk water are included in the Supplementary Information to demonstrate the generality of the MLPs and to facilitate the usage of the potentials. The MLPs are not applicable for gas water molecules or gas molecules adsorbed on surfaces.

We employed the Behler-Parrinello artificial neural network[36], and using the N2P2 code[53]. The committee model[39] with four individual MLPs was used to improve accuracy and provide uncertainty estimations.

We also constructed Δ-learning potentials[54] for fitting to the SCAN and the PBE functionals. We used 3090 configurations for the Δ-learning to get the SCAN MLP, although the learning curves suggest that even 20% of these are sufficient. the training and testing errors for energies are 0.24 and 0.28 meV/atom, and training and testing errors are 50 meVÅ$^{-1}$ and 49 meVÅ$^{-1}$ for the atomic force components, respectively. For the Δ-learning PBE MLP, we used 3226 configurations. The training and testing errors for energies are 0.38 and 0.40 meV/atom, respectively. The training and testing errors for atomic force components are 60 and 70 meVÅ$^{-1}$, respectively. The SCAN and the PBE Δ-learning MLPs are applicable to bulk water and the eight surfaces that are either pristine or with simple stoichiometric surface defects in water.

## Benchmark of the MLP

The accuracy of our MLPs was validated by the following benchmarks as detailed in the Supplementary Information: The predicted lattice constants of bulk anatase/rutile $TiO_2$ using the optB88-vdW MLPs agree well with the previous DFT calculations and experiments. The relaxed surface energies of eight pristine surfaces (anatase (001), (100), (101) and (110); rutile (001), (011), (100) and (110)) are in good agreement with our optB88-vdW DFT calculations as well as previous DFT results. Comparing the optB88-vdW DFT MD and optB88-vdW MLP MD simulations for the interfaces between water and various $TiO_2$ facets with and without surface defects, we get a good

agreement for the density profiles of the oxygen and hydrogen atoms, the oxygen-oxygen radial distribution functions of water molecules, and the orientation distribution of water on the surfaces. The water density profile based on our optB88-vdW MLP agrees well with Schran et al.[39] for rutile (110) using the same simulation setup, and our SCAN MLP water density profile agrees well with Andrade et al.[23] for anatase (101) with the SCAN functional. Moreover, for all the seven interfaces reported in Fig. 1, MLP and DFT energies and atomic forces at the optB88-vdW level agree well for configurations generated from the MLP metadynamics simulations.

## MLP MD simulation details

All MD simulations were performed in LAMMPS[55] with a MLP implementation[56]. The timestep is 1 fs throughout.

The metadynamics calculations of free energy surfaces of water dissociation were performed using LAMMPS[55] patched with the PLUMED code[57]. The PLUMED input file with the specification of the CV is provided in the SI repository. NVT simulations were used with Nosé-Hoover thermostat, with the fixed volume of the simulation box set such that the water density at the center kept at 1 gmL$^{-1}$. The cross-section of the simulation box is commensurate with the experimental lattice parameter of $TiO_2$. For using the PBE MLPs and SCAN MLPs, we used the hybrid pairstyle in LAMMPS in order to apply the original optB88-vdW MLP simultaneously with the Δ-learning potentials. We performed one independent metadynamics run for each MLP (3 DFT functional times 4 committee MLP models) and for each surface (8 surfaces). Each independent metadynamics run lasts 5 ns.

## Reporting summary

Further information on research design is available in the Nature Portfolio Reporting Summary linked to this article.

## Data availability

The machine learning potentials, training sets, sample DFT and metadynamics input files, PYTHON data analysis scripts and other necessary source data files generated for this study are available in the SI repository (https://github.com/BingqingCheng/TiO2-water)[58].

## Code availability

The MD simulations were performed using the LAMMPS code[55] with a MLP implementation[56]. The ASAP package is publicly available at https://github.com/BingqingCheng/ASAP[59].

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

## Acknowledgements

F.S., J.H., and B.C. thank the Swiss National Supercomputing Centre (CSCS) for the generous allocation of CPU hours via production project s1108 at the Piz Daint supercomputer. B.C. acknowledges resources provided by the Cambridge Tier-2 system operated by the University of Cambridge Research Computing Service funded by EPSRC Tier-2 capital grant EP/P020259/1. J.C. acknowledges the Beijing Natural Science Foundation for support under grant No. JQ22001. F.S., and J.H. thank the Swiss Platform for Advanced Scientific Computing (PASC) via the 2021-2024 "Ab Initio Molecular Dynamics at the Exa-Scale" project. This project has received funding from the European Union's Horizon 2020 research and innovation programme under the Marie Skłodowska-Curie grant agreement No 101034413.

## Author contributions

J.H., and B.C. conceived the idea; B.C. designed the research; Z.Z., F.W., K.L., F.S., and B.C. performed the research; Z.Z., F.W., K.L., F.S., J.H., J.C. and B.C. wrote the paper.

## Competing interests

The authors declare no competing interests.
