## [Peer Review File · Nature Communications]

REVIEWER COMMENTS

Reviewer #1 (Remarks to the Author):

This manuscript reports machine learning potential (MLP) simulations of water adsorption and dissociation on low-index rutile and anatase TiO₂ surfaces. The work extends recent MLP studies (refs. 14, 15, 23, 38) focused on the water-rutile (110) and water-anatase (101) interfaces by comparing both the behaviors of several different TiO₂ surfaces and the predictions of three different DFT functionals. Moreover, the authors were able to construct MLPs simultaneously applicable to many different interfaces and visualized their results using a machine learning method recently developed by one of the authors (ref. 47).

Overall, this is an interesting manuscript on the application of state-of-the-art machine learning techniques to systems of fundamental and technological interest. There are however several fundamental and technical concerns that the authors should address before this manuscript may be further considered for publication.

- The training set of the MLPs should also contain configurations near the transition states for proton transfer if the proton transition rates are to be accurately captured. The authors should discuss how this part of the free energy landscape was sampled and compare proton transfer potentials of mean force between MLP-MD and AIMD to understand if the barrier regions are properly captured.

- Given the known dependence of the water dissociation fraction on slab thickness for rutile (110) (see refs 12, 14, 15), the authors should also present results for another slab thickness in order to validate their MLP. In particular, their PBE results for this surface appear to favor water dissociation (see Fig. 1 and Table S2) in contrast to previous studies using these functionals (refs 12, 14). This difference should be discussed.

- The authors focus more on the methodological aspects of the work, but do not discuss the reason why the investigated surfaces exhibit different proton transfer mechanisms as well as why water dissociation on TiO₂ surfaces is important.

- The authors did not clearly explain how they assign the protons to different sites. They could also compare their method to that described in ref. [doi/10.1073/pnas.1819771116](https://doi.org/10.1073/pnas.1819771116)

- The Supplementary Information mentions results for defected surfaces (Figs. S1, S2, S3, S5, etc) that are never described or even mentioned in the main text. In fact, all results in the main text refer to pristine surfaces (see, e.g., Fig. 1). Similarly, the SI reports results for the anatase (001) surface, whereas in the main text it is pointed out that the FES for this surface lacks convergence, so it was removed from further analysis.

- DFT calculations were performed using a planewave energy cutoff of 350 Ry for both optB88-vdW and SCAN functionals. However, this cutoff is known to be far too low for SCAN. In fact, a much higher cutoff of 1,200 Ry was found necessary in recent studies (ref. 15; see also. 10.1021/acs.jctc.0c00300 and refs. 60, 61 therein)

- Fig. 3d showing proton transition probabilities is visually nice but difficult to understand in detail. The authors could present a detailed analysis of a specific example in the SI along with a Table reporting relevant values for all the different surfaces.

Reviewer #2 (Remarks to the Author):

The manuscript by Zheng et al. reports the simulation of water dissociation over different surfaces of the anatase and rutile TiO₂ polymorphs using ML potentials trained on DFT reference data. The topic has recently received renewed interest, also because of the wider availability of ML potentials. The manuscript is well written.

The main novelty of this work lies in the approach. The authors employed unsupervised learning and statistical techniques to compare the dissociation mechanisms on the different surface facets. The identified mechanisms are no new but are in agreement with previous reports. In addition to the relatively low novelty, some of the reported analyses (such as most of Fig. 2) do not actually seem to contribute to the work's narrative, and I have some concerns about the validation of the ML potentials.

1. ML potential benchmarks

The ML potential benchmarks reported in the SI do not instill confidence in the reader. First, tests for some of the most relevant quantities are not reported and should be added in a revision: How do the MLPs perform for (1) the relative energies of rutile and anatase, (2) the surface energies of the non-

polar surfaces, (3) the adsorption energies for H₂O, H, O, and OH on different sites, and (4) the energies of surface defects? Second, the various density profiles show significant differences between DFT and the MLP that are at least on the order of the features discussed in Fig. 1 of the main manuscript. One example is $\rho(\text{O})$ anatase-110-d-0. How can you be certain that the differences seen in Fig. 1 are quantitative and unrelated to errors in the ML potentials?

2. Asymmetric density profiles

Why are the density profiles shown in the SI asymmetric, i.e., different for the two sides of the symmetric slab models? Do these differences indicate a lack of convergence? Or are the slabs not actually symmetric? Are the asymmetries no longer present in the 5-ns long MLP-based MD simulations?

3. Presentation of the benchmarks

Related to the previous question: the colors and line types of the density plots make it challenging to discern DFT and MLP. I would recommend using different colors to distinguish between DFT and MLP and solid/dashed (not dotted/dashed) lines to distinguish between the two sides of the slab.

4. Short distances in water density profiles

The water density for the anatase (100) and (110) surfaces exhibit features at very short distances from the surface. Can these be explained, or are they artifacts of ML potential failures?

5. Data in Figure 2

Only subfigures a, d, and j of Figure 2 are discussed in the manuscript. If the other data is unnecessary, I recommend moving it to the SI. In addition, all symbols should be explained wherever the figures are shown.

6. ML potential availability

Please comment on whether the ML potential parameters will be made publicly available for others to use. This statement should be added to the "Data Availability" section.

Reviewer #3 (Remarks to the Author):

The manuscript presents a very interesting theoretical study of water dissociation at TiO₂/water interfaces. Despite the importance of TiO₂ in photocatalysis, such a fundamental system as the TiO₂/water interface still raises questions regarding the nature of the water adsorption at this interface. This study took a novel approach of using machine learning potentials trained on DFT results, to investigate the interfaces of several TiO₂ rutile and anatase surfaces with bulk water. The study presented a systematic picture of stabilities of dissociative vs molecular adsorption on the different TiO₂ surfaces, and clarified whether the observed stabilities are dependent on the surface structures or on the underlying DFT method. Further, the study analysed the mechanism of water dissociation and distinguished two types of mechanisms for water dissociation, which were at play on different surfaces.

This study gives new insights into the nature of TiO₂/water interfaces and the dynamics of water at these interfaces, and therefore makes an important contribution to the understanding of surface chemistry of TiO₂, with implications for photocatalysis, e.g. photocatalytic water purification and water splitting. The work is carefully performed and thoroughly analysed, and the results are placed in the context of the literature to date. Therefore I recommend publication in Nature Communications, after the following comments are addressed.

I have a question on the use of the PBE functional: although PBE is perhaps the most widely used DFT functional, it is known to underestimate weak interactions, and therefore is often used in conjunction with a dispersion correction. Why was dispersion correction not used here? This would probably have provided a fairer comparison with the van der Waals-corrected optB88-vdW and meta-GGA SCAN functionals.

Minor comments:

- p. 2: "On these surfaces, two water molecules are absorbed by surface undercoordinated four-(Ti4c) sites, and one molecule on five-fold (Ti5c) sites" – this sounds like all surfaces contain both Ti4c and Ti5c sites; needs re-phrasing.
- Fig. 2: what are the principal components of the kPCA map (the axes in Fig. 2)?
- "Absorption" and "absorbed" are sometimes used instead of "adsorption" and "adsorbed", this needs to be corrected throughout the manuscript.

We thank the referees for their careful reading of our manuscript. We have made a number of changes in response to their comments, and have highlighted these in blue in the revised version of the text. In what follows, we respond to each of the points raised by the referees.

Reviewer #1 (Remarks to the Author):

This manuscript reports machine learning potential (MLP) simulations of water adsorption and dissociation on low-index rutile and anatase TiO₂ surfaces. The work extends recent MLP studies (refs. 14, 15, 23, 38) focused on the water-rutile (110) and water-anatase (101) interfaces by comparing both the behaviors of several different TiO₂ surfaces and the predictions of three different DFT functionals. Moreover, the authors were able to construct MLPs simultaneously applicable to many different interfaces and visualized their results using a machine learning method recently developed by one of the authors (Ref. 47).

Overall, this is an interesting manuscript on the application of state-of-the-art machine learning techniques to systems of fundamental and technological interest. There are however several fundamental and technical concerns that the authors should address before this manuscript may be further considered for publication.

We thank the referee for examining our work and providing useful suggestions. We have extensively revised the study to address the technical concerns raised by the referee.

- The training set of the MLPs should also contain configurations near the transition states for proton transfer if the proton transition rates are to be accurately captured. The authors should discuss how this part of the free energy landscape was sampled and compare proton transfer potentials of mean force between MLP-MD and AIMD to understand if the barrier regions are properly captured.

Indeed, it is crucial for the training set of the MLP to contain configurations along transition paths in order to accurately describe the free energy profile of water dissociation. To do that, we iteratively collected configurations from metadynamics simulations: snapshots were selected from the trajectories generated by a previous generation of the MLP, and then recomputed using optB88-vdW DFT. We performed a total of 4 rounds of iterations. A total of 6688 configurations were selected in this way. We have added the description of the training set generation to the Method section of the manuscript.

The proton transfer potentials of mean force from AIMD are difficult to compute in this case due to the timescale limitations. We did, however, compare our FES to Ref.23, which used a MLP based on the SCAN functional, for the anatase (101) surface and reached good agreement.

Moreover, to estimate the uncertainty from the construction of the MLPs, we used committee models consisting of four individual MLPs. The spread in the FES, density profile predictions and water orientation distributions from the four MLPs is overall quite small. More importantly,

the uncertainties of the free energies near the transition states are similar with those of equilibrium states. This provides confidence that the MLP is accurate for the transition states.

- Given the known dependence of the water dissociation fraction on slab thickness for rutile (110) (see refs 12, 14, 15), the authors should also present results for another slab thickness in order to validate their MLP. In particular, their PBE results for this surface appear to favor water dissociation (see Fig. 1 and Table S2) in contrast to previous studies using these functionals (refs 12, 14). This difference should be discussed.

We thank the referee for raising this issue. We have thoroughly investigated the dependence of the water dissociation fraction on slab thickness, not just for rutile (110), but for all seven surfaces that we studied.

We used optb88-vdw MLP metadynamics simulations to compute the free energy surfaces (FES) of water dissociation (see Fig. R1 shown below).

For rutile (110), previous calculations (Ref.12, 14,15) predicted that water dissociation fraction has an odd-even oscillation behavior with respect to the number of layers (one layer is defined as one O-Ti-O trilayer for rutile (110) as shown in the inset of Fig. R1). We also observed the same odd-even oscillation for the free energy difference between water molecular adsorption and dissociation on rutile (110) surface, and the oscillation finally subsides after a slab thickness of around 10 layers. The fact that our MLP can reproduce the oscillation behavior at different rutile (110) slab thickness evidences the reliability and generality of our MLPs.

For the other six surfaces, no evident odd-even oscillation behavior is observed in the FES. For rutile (110), we re-computed the FES using a slab thickness of 10 trilayer, using optB88-vdW MLP, SCAN MLP and PBE MLP. The relevant results are updated in Fig.1, Fig.2 and Fig.3d, along with discussions in page 2 and 4 of the main text.

Fig. R1 Free energy surfaces with different slab thicknesses calculated from optB88-vdW MLP metadynamics simulations for seven surfaces. For rutile (110), we used the committee model with four individual MLP fits to estimate the standard deviations (shaded areas).

We have further discussed our results on rutile (110) by comparing with previous studies: At 10 layers, our converged free energy difference between dissociated water and molecular water is 5.2 ± 0.6 kJ/mol (with molecular state being more stable) at the optb88-vdw MLP level, 2.6 ± 1.2 kJ/mol at the SCAN MLP level, and 5.1 ± 0.6 kJ/mol at the PBE MLP level. Thus our results using 10 layers agree with previous studies that rutile (110) favors molecular adsorption.

Ref. 12 reported that water does not dissociate on rutile (110) surface using PBE AIMD simulations at 360 K with a simulation time of 40 ps, although water dissociation may be kinetically hindered within the short-time AIMD simulation due to the large activation barrier. Indeed, Selloni et al. (Ref. 23 in the main text) reported that neither the dissociation of water nor the recombination of hydroxyls were observed in AIMD simulations of the anatase (101) water interface of approximately 40 ps duration. Rutile (110) has an even higher free energy barrier of water dissociation/recombination than anatase (101), so even longer equilibration time may be needed.

In Ref. 14, Zhuang et al. performed PBE-D3 MLP MD simulations at 330 K, and revealed the odd-even oscillation of the water dissociation fraction on rutile (110) surface. For very thick slabs, they predicted a water dissociation rate of 2%, which corresponds to a free energy difference of about 10.7 kJ/mol between dissociatively and molecularly adsorbed water at the interface, which is larger than that of our results of 5.1 ± 0.6 kJ/mol from PBE MLP metadynamics simulations with 10 trilayers. The difference may be because: (i) There is a difference in the functional used (PBE and PBE-D3); (ii) We used a simulation temperature to 370 K compared to 330 K used in Ref. 14 as PBE water has a high melting point of about 417 K; (iii) We used metadynamics to reliably estimate the FES, while Ref. 14 used MD simulations.

Ref. 15 used SCAN MLP MD, and predicted the odd-even oscillation and a free energy difference of 3.9 ± 0.7 kJ/mol between dissociatively and molecularly adsorbed water at the interface. This agrees well with our results from SCAN MLP metadynamics simulations with a free energy difference of 2.6 ± 1.2 kJ/mol.

We have updated our discussion regarding water dissociation on rutile (110) in the main text (page 4 of the manuscript).

- The authors focus more on the methodological aspects of the work, but do not discuss the reason why the investigated surfaces exhibit different proton transfer mechanisms as well as why water dissociation on TiO₂ surfaces is important.

Different proton transfer and water dissociation mechanisms are related to the atomic arrangement on different TiO₂ surfaces: The one-step mechanism requires a close distance between surface O_{2c} and the protons in water adsorbed by surface Ti (H₂O-Ti), and the two-step mechanism needs the proximity of H₂O⁽¹⁾ to the surface.

For example, anatase (100) and (101) surfaces have adsorbed water ($\text{H}_2\text{O-Ti}$) with H atoms pointing upwards, so the distances between these hydrogen atoms and O_{2c} are relatively large, making direct water splitting difficult. Meanwhile, $\text{H}_2\text{O}^{(1)}$ are close to the surface, with many hydrogen atoms pointing downwards, facilitating the two-step proton transition and water dissociation mechanism. As such, both surfaces tend to have only the two-step mechanism. For the rutile (001), (011), and (100) facets, the water density profiles (Fig. 1b) have high first peaks due to $\text{H}_2\text{O-Ti}$, while the second peaks (contributed by $\text{H}_2\text{O}^{(1)}$) are far from the surfaces. In this case, the dense $\text{H}_2\text{O-Ti}$ layer and the far $\text{H}_2\text{O}^{(1)}$ layer make the one-step process favorable and the two-step process unlikely.

For rutile (110), both conditions (closeness between surface O_{2c} and protons in $\text{H}_2\text{O-Ti}$, proximity of $\text{H}_2\text{O}^{(1)}$ to the surface) are satisfied, leading to the coexistence of the two mechanisms. We have expanded our discussions of the mechanisms in the main text to make these points clearer.

We have also expanded the discussion on the implications of our results in the Conclusion section of the manuscript. Our findings about water dissociation on TiO_2 surfaces are important for the following reasons:

(i) Water dissociation fraction, free energy barrier and proton transfer on surfaces are key features for investigating the reactivity of TiO_2 -water interfaces in chemical or photochemical settings, which is relevant for numerous practical applications. Previous studies almost exclusively focus on the anatase (101) and rutile (110) surfaces, and we comprehensively elucidate water adsorption and dissociation processes on seven low-index surfaces in aqueous environments for the first time. We show that different pristine TiO_2 surfaces react with water in distinct ways, and cannot be represented using just the low-energy anatase (101) and rutile (110) surfaces. Surfaces such as anatase (100), (110) and rutile (001), (011) may be more reactive in photochemical water splitting than the stable surfaces as they favor more water dissociations. Our results thus imply that, in order to better understand the photocatalysis, catalysis and biomedical applications of TiO_2 (nano)particles, the high-energy surfaces need to be taken into account.

(ii) We rationalized the different water dissociation and proton transfer mechanisms on different surfaces based on the atomic arrangements. This is not only crucial for understanding the water interaction with these pristine interfaces, but also paves the way towards more complex surfaces with defects, polarons and reconstructions.

(iii) Elucidating the water dissociation/proton transfer rate and mechanisms of the TiO_2 -water interface is essential for improving the performance of TiO_2 in photocatalysis applications and can also help understand and improve the functional properties of other oxide materials operating in aqueous environment.

- The authors did not clearly explain how they assign the protons to different sites. They could also compare their method to that described in Ref. doi/10.1073/pnas.1819771116

We have now expanded our explanation on how to assign the protons to different sites. The kPCA maps are constructed from a comprehensive set of features characterizing the environment of hydrogen atoms in the system. From the kPCA maps in Fig.2, the whole set of H environments forms well-separated clusters (rather than having a continuous spectrum of environments), suggesting that it is physically meaningful to assign protons to different sites. As the clusters are distinct, we found that a few assignment schemes all work quite well, such as automatic clustering methods (e.g. DBSCAN) and decision-tree methods. Eventually we used a simple decision tree based on the following criteria:

- If the hydrogen distance to its nearest oxygen in TiO_2 is more than 5 Å, it is classified to be far from the surface ($\text{H}_2\text{O}^{(>1)}$)
- Instead if the hydrogen distance to its nearest oxygen in TiO_2 is less than 1.25 Å, and the hydrogen distance to its closest neighboring hydrogen (H-H) is more than 1.6 Å, the hydrogen is classified as H adsorbed on the surface O (H-O_t)
- Else if $\text{H-O}_t > 1.85$ Å, and the nearest O_w -Ti distance is less than 3 Å, this is a OH adsorbed on Ti (HO-Ti)
- Else if the nearest O_w -Ti distance is less than 3 Å, it is adsorbed H_2O ($\text{H}_2\text{O-Ti}$).
- Else the hydrogen is classified as first-layer H_2O ($\text{H}_2\text{O}^{(1)}$).

The same criteria were used for all surfaces, and the resulting classifications are illustrated in Fig.9,10 of the SI. The classification is not very sensitive to the cutoffs used above, as long as they are within a reasonable range.

We have expanded the explanations in the main text and in the SI of the paper.

The paper (PNAS 116 (10) 4054-4057) mentioned by the referee provides a Voronoi-constellation-based method to assign protons to different acid-base sites. We think the overall idea can be potentially combined with the workflow that we used for the analysis, but some extensions are needed because we need to 1) classify H in neutral water, and 2) distinguish the locations of protons in relation to the surface sites.

- The Supplementary Information mentions results for defected surfaces (Figs. S1, S2, S3, S5, etc) that are never described or even mentioned in the main text. In fact, all results in the main text refer to pristine surfaces (see, e.g., Fig. 1). Similarly, the SI reports results for the anatase (001) surface, whereas in the main text it is pointed out that the FES for this surface lacks convergence, so it was removed from further analysis.

The MLPs for TiO_2 /water that we constructed are applicable to both pristine and defective stoichiometric surfaces. Our vision is to ultimately model realistic aqueous surfaces (with defects). As an initial and important step, in this paper we develop methods and study high-energy pristine surfaces. This is not only crucial for paving the way to investigate defected surfaces, but also help resolve experimental/theoretical controversies about water dissociation on pristine surfaces.

Although this paper does not describe results for defective surfaces, we decided to include the benchmarks on these surfaces in the SI for the following reasons: (i) To facilitate the usage of these MLPs by other groups. In the section of Methods for the MLP construction, we provided instructions for the applicable scope of our MLP on defective surfaces. (ii) To showcase the accuracy of the MLPs even for defected surfaces, which indirectly help prove the generalizability of the potential.

We have now added a sentence in the Method section to explain why we included the benchmarks for the defected surfaces.

The FES for the anatase (001) surface is removed from further analysis in the main text, because the choice of collective variable is not optimal in the metadynamics simulations so the calculated FES lacks convergence. This is not an issue of the MLPs - the MLPs can reliably compute other properties such as surface energy and water density profile of anatase (001). We thus kept the benchmark results for anatase (001) in the SI.

- DFT calculations were performed using a planewave energy cutoff of 350 Ry for both optB88-vdW and SCAN functionals. However, this cutoff is known to be far too low for SCAN. In fact, a much higher cutoff of 1,200 Ry was found necessary in recent studies (Ref. 15; see also. 10.1021/acs.jctc.0c00300 and refs. 60, 61 therein)

We thank the referee for raising the issue of the planewave energy cutoff for SCAN. We have thoroughly investigated this issue, recomputed the training set of the SCAN MLP using 1200 Ry, and updated all the relevant results.

First, we performed a convergence test based on a small set of TiO₂-water configurations using a range of different planewave cutoffs. In the figures below we compare the energies and the forces computed using a 1200 Ry cutoff (x-axes) and 350 Ry, 600 Ry, 800 Ry, 1000 Ry cutoff values (y-axes). Although the energies are already well-converged at 350 Ry cutoff, the forces on atoms are slower to converge with respect to the cutoff. Our benchmark is consistent with the remark in <https://pubs.acs.org/doi/pdf/10.1021/acs.jctc.7b00846>, noting that 1200 Ry was used to ensure that the forces are well-converged. These benchmark results are added to the SI.

We then recomputed the training set of the SCAN MLP using the 1200 Ry cutoff, and refitted the potentials. We noticed that the training errors went down dramatically and are currently similar with the errors of the PBE MLP. Before, using the training set using the 350 Ry cutoff, the training and testing errors for energies are 0.47 and 0.52 meV/atom, respectively, and training and testing errors are both 290 meV/Å for the atomic force components. With the new training set computed using 1200 Ry cutoff, the training and testing errors for energies are 0.24 and 0.28 meV/atom, and training and testing errors are 50 meV/Å and 49 meV/Å for the atomic force components, respectively.

We have rerun the MD simulations using the new fits of the SCAN MLPs, and updated the results in Fig.1 accordingly.

Actually, all the results, including the free energy profiles, density profiles and water orientations, do not change significantly. This is probably because, previously, the MLPs were taking care of the noise in the training set computed by SCAN with 350 Ry cutoff by acting as an interpolator between different atomic configurations. As such, although the old SCAN training set has larger noises on forces, the trained MLPs are very similar to the updated ones.

- Fig. 3d showing proton transition probabilities is visually nice but difficult to understand in detail. The authors could present a detailed analysis of a specific example in the SI along with a Table reporting relevant values for all the different surfaces.

We added detailed explanations for Fig. 3d by using anatase (101) as an example in the SI, and also added a Table (see page 16, Table S4 and S5 in the SI) to report specific values of the H transition probabilities between different states for all seven surfaces.

The following is a detailed analysis of Fig. 3d we added to SI:

We use anatase (101) as an example to explain the proton transition matrix. The matrix has a dimension of 5 x 5 as we have classified all hydrogen environments into five states. Since transitions of protons between certain states are not possible, the corresponding matrix elements remain unfilled with color (the probability is equal to zero). For instance, protons from $\text{H}_2\text{O}^{(>1)}$ to surface FF (H-O_t). For the one-step mechanism, $\text{H}_2\text{O-Ti}$ and H-O_t are the two involved states, and the corresponding element is marked as a red star in the lower triangle of the matrix. From Fig. 3d, we see that this transition probability is zero, indicating that the one-step mechanism is absent on anatase (101) surface as H atoms in H-O_t cannot proton-transferred from $\text{H}_2\text{O-Ti}$. In contrast, H can be transferred from the $\text{H}_2\text{O}^{(1)}$ to H-O_t (see the element marked by cyan triangle in Fig. 3d). The two-step mechanism ($\text{H}_2\text{O}^{(1)}$ and H-O_t) thus dominates the proton transport and water dissociation on anatase (101) surface. This result also agrees well with the individual proton transfer pathways shown in Fig. 3b of the manuscript.

Reviewer #2 (Remarks to the Author):

The manuscript by Cheng et al. reports the simulation of water dissociation over different surfaces of the anatase and rutile TiO_2 polymorphs using ML potentials trained on DFT reference data. The topic has recently received renewed interest, also because of the wider availability of ML potentials. The manuscript is well written.

The main novelty of this work lies in the approach. The authors employed unsupervised learning and statistical techniques to compare the dissociation mechanisms on the different surface facets. The identified mechanisms are no new but are in agreement with previous reports. In addition to the relatively low novelty, some of the reported analyses (such as most of Fig. 2) do not actually seem to contribute to the work's narrative, and I have some concerns about the validation of the ML potentials.

We thank the referee for the helpful suggestions for our work.

In our work, the water dissociation free energies and mechanisms for the two well-studied surfaces (anatase (101) and rutile (110)) are indeed in agreement with previous reports, as the referee noted. Such agreement supports the validity of our MLP and our methods. However, the water adsorption and dissociation on other high-energy surfaces was not explored at all in previous simulation studies. We think the novelty of our manuscript mainly comes from the investigation of all major low-index surfaces of TiO_2 under the same setting. This enables a more holistic and physical understanding of water reactions on TiO_2 particles that involves different surfaces. We show that different pristine TiO_2 surfaces react with water in distinct ways, and cannot be represented using just the low-energy anatase (101) and rutile (110) surfaces. For example, anatase (100), (110) and rutile (001), (011) may be more reactive in photochemical water splitting than the stable surfaces as they favor more water dissociations.

This implies that, in order to better understand the photocatalysis, catalysis and biomedical applications of TiO₂ (nano)particles, the high-energy surfaces need to be taken into account.

Moreover, to study the water dissociation mechanism for many surfaces we developed an automated workflow, which will facilitate future studies for defective surfaces, water on other materials, and complex aqueous solutions.

In the revision, we have highlighted the novelty of our work, expanded the discussion on the implications, and added new validation of the MLPs.

1. ML potential benchmarks

The ML potential benchmarks reported in the SI do not instill confidence in the reader. First, tests for some of the most relevant quantities are not reported and should be added in a revision: How do the MLPs perform for (1) the relative energies of rutile and anatase, (2) the surface energies of the non-polar surfaces, (3) the adsorption energies for H₂O, H, O, and OH on different sites, and (4) the energies of surface defects? Second, the various density profiles show significant differences between DFT and the MLP that are at least on the order of the features discussed in Fig. 1 of the main manuscript. One example is $\rho(O)$ anatase-110-d-0. How can you be certain that the differences seen in Fig. 1 are quantitative and unrelated to errors in the ML potentials?

We have added additional benchmarks for the performance of the MLPs:

(1) The optimized energies for anatase and rutile TiO₂ at 0 K calculated by optB88-vdW MLP are shown in Table R1.

Table R1: Relaxed energies of anatase and rutile phases at 0 K calculated using optB88-vdW MLP with committee model. The errors of the MLP values are from the standard deviations of the estimates from the 4 individual fits.

Phase	Energy (eV/atom)
Anatase	-822.813 ± 0.001
Rutile	-822.788 ± 0.004

Anatase is found to be more stable than rutile by 0.025 eV/atom at the optB88-vdW MLP level. This result contradicts experiments, but it is well-known that DFT calculations and experiments give contradicting results regarding the stability of the two phases. Our result is consistent with previous DFT calculations that predicted anatase to be more stable than rutile at 0 K (e.g., J. Phys.: Condens. Matter 24 405501 (0.032 eV/atom at the PBE functional level); PhysRevB.63.155409 (0.033 eV/atom at the PBE functional level, 0.01 eV/atom at the LDA functional level, and 0.04 eV/atom at the BLYP functional level) and J. Chem. Phys. 156,

074106 (2022) (0.021 eV/atom at the PBEsol functional level)). Our previous study (Phys. Chem. Chem. Phys., 2020, 22, 12697) also found anatase to have lower lattice energy than rutile at 0 K and ambient pressure, using DFT calculations based on LDA (0.014 eV/atom), PBE (0.048 eV/atom) and PBEsol (0.021 eV/atom).

We have included the computed lattice energies of the anatase and rutile phases in the SI, along with relevant discussions.

(2) We computed the surface energies (SE) of all eight surfaces (anatase (001), (100), (101) and (110), rutile (001), (011), (100) and (110)) at 0 K. The surfaces were relaxed at the MLP level.

Table R2: SE of eight surfaces at 0 K calculated using optB88-vdw MLP with the committee model. The errors of the MLP values are from the standard deviations of the estimates from the 4 individual fits.

Surfaces	Surface energy (J/m ²)
anatase (001)	1.327 ± 0.006
anatase (100)	0.881 ± 0.006
anatase (101)	0.680 ± 0.005
anatase (110)	1.421 ± 0.006
rutile (001)	1.325 ± 0.002
rutile (011)	1.174 ± 0.003
rutile (100)	0.912 ± 0.006
rutile (110)	0.811 ± 0.004

Previous DFT calculations at 0 K by Lazzeri et al. (PhysRevB.63.155409) and Barnard et al (PhysRevB.70.235403) both reported that the values of SE of anatase TiO₂ follow the sequence (101) < (100) < (001) < (110), and anatase (101) owns the minimum SE (~ 0.52 J/m² within PBE functional (PhysRevB.63.155409) and ~0.85 J/m² within LDA functional (J. Chem. Theory Comput. 2008, 4, 2, 341–352), which both agree well with our results.

For rutile TiO₂, our calculations indicate that the (110) surface has the smallest SE among the four surfaces at 0 K, and it agrees well with previous DFT calculations based on PW91 (0.68 J/m²) (J. Phys.: Condens. Matter 18 4207) and PBE (0.74 J/m²) (Applied Surface Science 436 (2018) 989–9) functionals. At 0 K, the ordering of SE is (110) < (100) < (011) < (001), also in agreement with previous DFT results from Ramamoorthy et al. (Phys. Rev. B 49, 16721).

We have added the SE results and the comparisons into the SI (Table S2).

(3) Our training set does not include the configurations for isolated adsorbed water (gas-phase H₂O) on anatase/rutile surfaces. Therefore, this benchmark lies outside the scope of application for our MLPs. The primary focus of our study is to elucidate the TiO₂-water interactions under aqueous conditions, thus we did not investigate the adsorption and dissociation of gas-phase H₂O on the surfaces. We have added a sentence in the Methods section to explain the applicable range of the MLPs.

(4) This work mainly focuses on studying the water adsorption and dissociation on the pristine low-index TiO₂ surfaces, so the energies of surface defects are not relevant. Moreover, there are many different surface defects (e.g, step edge, kinks, polarons and vacancies) on TiO₂ surfaces in experiments, so it is not straightforward to decide which defects to include in the benchmark.

The referee also pointed out the difference between density profiles based on DFT and the MLP. This is due to the limitations on the timescale of AIMD simulations and the slow convergence of the density profiles, as we further explain in the response below. For the sake of benchmarking the MLPs, the simulation setup of the MLP MD was selected to closely resemble the AIMD runs: same initial configurations, same thermodynamic conditions, same time step size and simulation length. As such, we are comparing apples with apples - similarly unequilibrated trajectories from MLP and DFT. Considering this, the agreement between the two is very good. Of course, such comparison is less rigorous than comparing the equilibrated density profiles, but unfortunately such quantities from nanosecond-long AIMD is not affordable.

Moreover, to estimate the uncertainty from the construction of the MLPs, we used a committee consisting of four individual MLPs. The difference between the predictions of the different fits of the MLPs in the committee, as measured by the standard deviation of the predictions of the members, provides an estimate of the error of the MLPs. In this case, the spread in the FES, density profile predictions and water orientation distributions from the four individual MLPs (See Fig.1) is overall quite small. This suggests that our MLPs are reliable in modeling water dissociation and adsorption on the TiO₂ surfaces.

2. Asymmetric density profiles

Why are the density profiles shown in the SI asymmetric, i.e., different for the two sides of the symmetric slab models? Do these differences indicate a lack of convergence? Or are the slabs not actually symmetric? Are the asymmetries no longer present in the 5-ns long MLP-based MD simulations?

For the defect-free surfaces (d-0 in Fig. S2-S4), the two sides are symmetric. Indeed, the right-hand-sides of the slabs with defects (d-1 and d-4) are also constructed to be the same as the defect-free surfaces. In the 15ps AIMD or MLP MD simulations, because of the lack of time

convergence, the density profiles of the symmetric slabs appear asymmetric. The same convergence issue and the asymmetric density profiles from short simulations were also observed in Ref. 23 (Chem. Sci., 2020, 11, 2335). One key reason for the slow equilibration of the TiO₂-water system is due to the high free energy barrier of water dissociation (~10 kbT, as discussed in the main text). Another reason is the slow diffusivity of water near the surface, as discussed in Ref. 38 (PNAS2021 Vol. 118 No. 38). The asymmetries indeed diminish in long-time MD simulations, e.g. in the 5-ns MLP MD simulations of anatase (101) as shown in Fig. S7 of SI (reproduced below).

3. Presentation of the benchmarks

Related to the previous question: the colors and line types of the density plots make it challenging to discern DFT and MLP. I would recommend using different colors to distinguish between DFT and MLP and solid/dashed (not dotted/dashed) lines to distinguish between the two sides of the slab.

We have tried to revise these figures based on the referee's suggestion. We show an example of the revised figure below. However, we think the revised figures look a bit confusing as eyes tend to match curves with the same color. As such, we decided to use the original layout of the figures.

4. Short distances in water density profiles

The water density for the anatase (100) and (110) surfaces exhibit features at very short distances from the surface. Can these be explained, or are they artifacts of ML potential failures?

Such features are due to the fact that hydrogen atoms can be adsorbed by O_{2c} on these surfaces (see Fig. 1a in the manuscript). The adsorbed protons lead to the small bumps in the water density profiles very close to the surfaces. This phenomenon is observed in anatase (100), (110), and rutile (001), all of which favor water dissociation. We have added a sentence on page 2 of the main text to explain this.

5. Data in Figure 2

Only subfigures a, d, and j of Figure 2 are discussed in the manuscript. If the other data is unnecessary, I recommend moving it to the SI. In addition, all symbols should be explained wherever the figures are shown.

We have simplified Fig. 2 by removing most panels, and moved the rest of the data to the SI. We have also expanded the explanation in page 5 of the manuscript and defined all the symbols.

6. ML potential availability

Please comment on whether the ML potential parameters will be made publicly available for others to use. This statement should be added to the "Data Availability" section.

We have uploaded the MLPs and the MD input files to the publicly available Github repository: <https://github.com/BingqingCheng/TiO2-water>.

The training set of the MLPs is quite large, and we will upload it to public repositories such as MaterialsCloud upon the acceptance of the paper.

We have revised the Data Availability section as the following:

The machine learning potentials and other necessary input files generated for this study are available in the SI repository (<https://github.com/BingqingCheng/TiO2-water>).

Reviewer #3 (Remarks to the Author):

The manuscript presents a very interesting theoretical study of water dissociation at TiO₂/water interfaces. Despite the importance of TiO₂ in photocatalysis, such a fundamental system as the TiO₂/water interface still raises questions regarding the nature of the water adsorption at this interface. This study took a novel approach of using machine learning potentials trained on DFT results, to investigate the interfaces of several TiO₂ rutile and anatase surfaces with bulk water. The study presented a systematic picture of stabilities of dissociative vs molecular adsorption on the different TiO₂ surfaces, and clarified whether the observed stabilities are dependent on the surface structures or on the underlying DFT method. Further, the study analyzed the mechanism of water dissociation and distinguished two types of mechanisms for water dissociation, which were at play on different surfaces.

This study gives new insights into the nature of TiO₂/water interfaces and the dynamics of water at these interfaces, and therefore makes an important contribution to the understanding of surface chemistry of TiO₂, with implications for photocatalysis, e.g. photocatalytic water purification and water splitting. The work is carefully performed and thoroughly analyzed, and the results are placed in the context of the literature to date. Therefore I recommend publication in Nature Communications, after the following comments are addressed.

We thank the referee for the positive assessment of our work.

I have a question on the use of the PBE functional: although PBE is perhaps the most widely used DFT functional, it is known to underestimate weak interactions, and therefore is often used in conjunction with a dispersion correction. Why was dispersion correction not used here? This would probably have provided a fairer comparison with the van der Waals-corrected optB88-vdW and meta-GGA SCAN functionals.

The referee correctly points out that PBE underestimates weak interactions. Plain PBE water is known to exhibit issues including over-structuring in the oxygen-oxygen radial distribution functions, too high melting point, liquid water density too low, and ice being heavier than liquid water. Plain PBE is thus probably not amongst the best functionals for TiO₂-water. However, it is chosen in this study not because of its performance, but because we want to showcase and compare the predictions of distinct functionals for water dissociation on TiO₂, and also because there are many previous studies on TiO₂ and water using PBE to compare to.

We selected three sets of functionals, optB88-vdW and SCAN, and the plain PBE. The rationale for selecting the three sets is that they are all widely used in the previous studies of TiO₂-water and other aqueous systems, and they are distinct to each other. optB88-vdW is van der Waals-corrected, SCAN is meta-GGA and includes some van der Waals interaction, while PBE probably lacks most dispersion amongst the three. Despite the large difference in the construction and the performance of the three functionals in general, the predictions on the free energies of water dissociation on different TiO₂ surfaces are quite consistent. This validates the

robustness of our conclusions, and helps consolidate the previous literature on TiO_2 -water that employed different functionals.

Minor comments:

- p. 2: “On these surfaces, two water molecules are absorbed by surface undercoordinated four-(Ti4c) sites, and one molecule on five-fold (Ti5c) sites” – this sounds like all surfaces contain both Ti4c and Ti5c sites; needs re-phrasing.

We revised the sentence as the following:

On anatase (110) and rutile (001), two water molecules are adsorbed simultaneously by each surface undercoordinated four-fold (Ti_{4c}) site. For the remaining five surfaces, one water molecule is adsorbed on each five-fold (Ti_{5c}) site.

- Fig. 2: what are the principal components of the kPCA map (the axes in Fig. 2)?

The kPCA maps in Fig.2 show the similarity between different hydrogen atomic environments: similar H environments are close on the map and vice versa. The principal components capture the largest and the most important variance of the atomic environments. Mathematically, these principal components correspond to the largest eigenvectors of the kernel matrix. We have added explanation and reference in the main text.

- “Absorption” and “absorbed” are sometimes used instead of “adsorption” and “adsorbed”, this needs to be corrected throughout the manuscript.

We appreciate the referee for noticing this issue. We now use “adsorption” and “adsorbed” throughout the manuscript and SI.

REVIEWER COMMENTS

Reviewer #1 (Remarks to the Author):

The authors have substantially improved their manuscript by adding numerous benchmarks and expanding their discussion. While I am largely satisfied with this revised manuscript, I believe they could still improve the assessment of their FES curves (Figure 1d) by directly comparing the results of their MLPs to DFT on a small unit cell using configurations generated by MLP-based enhanced sampling simulations. They could do that on just a few of the interfaces they have studied and were not considered in previous work.

Reviewer #2 (Remarks to the Author):

The authors have addressed most of my concerns, except for the comparison of key materials properties with the DFT reference method.

Instead of reporting uncertainties from the MLP ensemble, a comparison of the MLP-predicted surface energies with the DFT surface energies calculated using the reference method (e.g., optB88-vdw) should be included. Such a comparison is needed to determine (1) how reliable the uncertainty estimate is and (2) how well the simulation results approximate the expected results of DFT-based simulations with the reference method. Similarly, the adsorption energy of a water monolayer on the different facets, as predicted with the MLP and the DFT reference method, could be reported if this data is available. Surely, this interaction is within the scope of the present work.

The total anatase/rutile energies are not meaningful and probably do not need to be reported. Though, the energy difference between the two phases as predicted by the MLP compared to that predicted by the concrete DFT reference method would inform the reader how well the MLP reproduces the reference method. Of course, the MLP will ideally exhibit the same order of phase stability as the reference method, which may or may not agree with nature. (The thermodynamic ground state of TiO₂ at zero Kelvin is still controversial, and quantum Monte Carlo also seems to predict that anatase is more stable than rutile.)

The comparison of the MLP results with other DFT methods, discussed in detail in the rebuttal letter, is interesting but ultimately does not provide a benchmark for the MLP.

As a remark, the additional explanation of the asymmetric density profiles indicates that the results are not converged and depend on the initial configuration due to an insufficient simulation time. I am therefore not convinced that the comparison of the MLP and DFT density profiles is at all meaningful. As the authors show, long MLP-based MD simulations give rise to a symmetric equilibrium density profile, but a comparison with DFT is not possible because of the long required simulation time.

Reviewer #3 (Remarks to the Author):

The authors have satisfactorily addressed my comments (and the other reviewers' comments). I recommend publication of this manuscript in Nature Communications.

We thank the referees for their comments. We have made a number of changes in response to their comments, and have highlighted these in blue in the revised version of the text. In what follows, we respond to each of the points raised by the referees.

Reviewer #1 (Remarks to the Author):

The authors have substantially improved their manuscript by adding numerous benchmarks and expanding their discussion. While I am largely satisfied with this revised manuscript, I believe they could still improve the assessment of their FES curves (Figure 1d) by directly comparing the results of their MLPs to DFT on a small unit cell using configurations generated by MLP-based enhanced sampling simulations. They could do that on just a few of the interfaces they have studied and were not considered in previous work.

We appreciate the referee's positive comments regarding our previous revision. Here we further made extensive comparisons between the MLP and DFT energies and atomic forces using configurations generated by MLP-based enhanced sampling simulations, as the referee suggested.

From each TiO_2 -water interface system (anatase (100), (101) and (110); rutile (001), (011), (100) and (110)), we first randomly selected configurations from MLP metadynamics simulations. We then performed single-point calculations using DFT based on the optB88-vdW functional using an energy cutoff of 350 Rydberg and were used for single-point DFT calculations. We finally compared the MLP and the DFT energy/forces for these configurations for each interface system, and plotted the results in Fig. S3 of the SI. These plots are reproduced below. Note that in these plots each data point is colored according to the collective variable (CV) $S_{\text{O-H}}$ used in the metadynamics simulations. The range of the CV for these configurations covers both the dissociated water and the molecular water range, which means that our selected configurations cover the whole water dissociation process.

For the energy comparison (see Fig. R1 below), we see that the raw data extracted from the DFT calculations exhibit a small energy offset compared to that from MLP calculations. Overall, the MLP overpredicts the stability of anatase. This offset could be attributed to the smaller system size and TiO_2 /water ratio used in the training set. Note that the energy offset for each interface system does not affect the free energy surface or the dynamics of the water adsorption and dissociation process. We also show comparison after adjusting for this offset, and observe excellent agreements between MLP and DFT energies. We reported the root mean square error (RMSE) of energy among all seven surfaces in the figures. The maximum RMSE after adjusting the energy offset is from the anatase (100) surface, which is found to be 0.78 meV/atom. Notably, this value is still smaller than the training RMSE of our MLP, which stands at 1.5 meV/atom.

The atomic forces also show good agreement: when examining the atomic forces (Fig. R1c), we find that the maximum RMSE among these seven surfaces is 103.82 meV/Å (again for the anatase (100) surface), which is also lower than the MLP's training RMSE of 133 meV/Å.

We performed an additional set of checks: we performed DFT calculations using the optB88-vdW functional on configurations from the anatase (100) surface, raising the energy cutoff from 350 to 600 Rydberg. Comparing these results (see Fig. R2 below), we see that the RMSEs of the energy and atomic forces remain largely unchanged with the increased energy cutoff in DFT calculations, demonstrating that an energy cutoff of 350 Rydberg is sufficient for accurately predicting the system's properties.

In summary, these extensive comparisons provide strong evidence of the reliability and accuracy of our MLP in precisely assessing the FES and water dissociation process. We added this benchmark into the Supporting Information. We also mention these comparisons in the Methods section of the main text.

Fig. R1 (a) Raw and (b) adjusted energy comparisons of randomly selected 100 configurations for seven TiO_2 surfaces calculated separately from DFT (optB88-vdW functional with energy cutoff of 350 Rydberg) and optB88-vdW MLP. Each point is color-coded based on the collective variable of the configuration it represents. (c) Atomic forces comparison of the same configurations for seven TiO_2 surfaces calculated separately using DFT and MLP.

Fig. R2 The comparison of energies and atomic forces of selected 100 configurations of anatase (100) interface system calculated separately from DFT (optB88-vdW functional) and optB88-vdW MLP. (a-c) An energy cutoff of 350 Rydberg was used for DFT calculations. (d-f) An energy cutoff of 600 Rydberg was used for DFT calculations.

Reviewer #2 (Remarks to the Author):

The authors have addressed most of my concerns, except for the comparison of key materials properties with the DFT reference method.

We appreciate the reviewer for the positive assessment on our previous revision. Now we added more comparison of key materials properties with the DFT reference method.

Instead of reporting uncertainties from the MLP ensemble, a comparison of the MLP-predicted surface energies with the DFT surface energies calculated using the reference method (e.g., optB88-vdw) should be included. Such a comparison is needed to determine (1) how reliable the uncertainty estimate is and (2) how well the simulation results approximate the expected results of DFT-based simulations with the reference method. Similarly, the adsorption energy of a water monolayer on the different facets, as predicted with the MLP and the DFT reference method, could be reported if this data is available. Surely, this interaction is within the scope of the present work.

We performed a direct comparison for the surface energies calculated from optB88-vdw DFT and optB88-vdw MLP. We computed the surface energies of eight surfaces (anatase (001), (100), (101) and (110); rutile (001), (011), (100) and (110)) at 0 K based on DFT using optB88-vdW functional and an energy cutoff of 350 Rydberg, and made a comparison (Table R1 below, and Table S2 In the SI) with those calculated from optB88-vdw MLP. We see that the surface energies of eight surfaces computed from MLP can reproduce the results of DFT very well, demonstrating the accuracy and reliability of our MLP.

Since the training data does not contain the water monolayer or monomer on TiO₂ interfaces, we cannot use it to calculate the adsorption energy of this particular process. Our MLP can be used to simulate the interactions between bulk water and TiO₂ surfaces. In order to further demonstrate our MLP can reliably access the water adsorption and dissociation processes on the interfaces, we randomly selected 100 configurations (cover both water adsorption and dissociation processes) from metadynamics simulations for each interface system, and then we used separately DFT and MLP to compute the energies and atomic forces of these configuration to make a direct comparison (see the reply for the question from referee 1 above).

Table R2: Surface energies of eight surfaces at 0 K calculated using separately DFT (optB88-vdW functional with an energy cutoff of 350 Rydberg) and optB88-vdW MLP with the committee model. The errors of the MLP values are from the standard deviations of the estimates from the 4 individual fits.

Surfaces	MLP surface energy (J/m ²)	DFT surface energy (J/m ²)
anatase (001)	1.327 ± 0.006	1.306
anatase (100)	0.881 ± 0.006	0.876
anatase (101)	0.680 ± 0.005	0.763
anatase (110)	1.421 ± 0.006	1.374
rutile (001)	1.325 ± 0.002	1.353
rutile (011)	1.174 ± 0.003	1.236
rutile (100)	0.912 ± 0.006	0.933
rutile (110)	0.811 ± 0.004	0.819

The total anatase/rutile energies are not meaningful and probably do not need to be reported. Though, the energy difference between the two phases as predicted by the MLP compared to that predicted by the concrete DFT reference method would inform the reader how well the MLP reproduces the reference method. Of course, the MLP will ideally exhibit the same order of phase stability as the reference method, which may or may not agree with nature. (The thermodynamic ground state of TiO₂ at zero Kelvin is still controversial, and quantum Monte Carlo also seems to predict that anatase is more stable than rutile.) The comparison of the MLP results with other DFT methods, discussed in detail in the rebuttal letter, is interesting but ultimately does not provide a benchmark for the MLP.

We thank the reviewer for this suggestion. We performed the DFT calculations and also found that anatase is slightly more stable than rutile by 0.67 meV/atom at the optB88-vdW level, which is smaller than the value at the MLP level but still consistent with the predicted order of phase stability from our MLP. We agree that the phase stability of anatase/rutile is controversial, and we highlighted this controversy in the Supporting Information.

As a remark, the additional explanation of the asymmetric density profiles indicates that the results are not converged and depend on the initial configuration due to insufficient simulation time. I am therefore not convinced that the comparison of the MLP and DFT density profiles is at all meaningful. As the authors show, long MLP-based MD simulations give rise to a symmetric equilibrium density profile, but a comparison with DFT is not possible because of the long required simulation time.

We thank the referee for the remark. We agree that the AIMD results are not thermodynamically converged due to the significant computational time required for DFT calculations spanning several nanoseconds, so we were unable to directly compare the converged density profiles obtained from DFT and MLP calculations. We have now highlighted this point in the SI. Nevertheless, we compared the DFT and the MLP density profiles ensuring that both simulations were performed under identical conditions in terms of simulation time duration, system size, initial configuration, and thermodynamic conditions. Under this setup, our results (Fig. S3-S5 in the Supporting Information) demonstrates that the density profiles generated by MLP calculations closely reproduce those obtained from DFT calculations. The results are not converged - but the kinetic effects partially cancel out due to the identical conditions.

Reviewer #3 (Remarks to the Author):

The authors have satisfactorily addressed my comments (and the other reviewers' comments). I recommend publication of this manuscript in Nature Communications.

We thank the reviewer for the positive assessments and recommendation.

REVIEWERS' COMMENTS

Reviewer #1 (Remarks to the Author):

The authors responses and revisions to the manuscript with regards to my concerns are satisfactory. I think this paper can be accepted for publication.

Reviewer #2 (Remarks to the Author):

In their second revision, the authors have addressed all of my remaining concerns. I can now recommend the publication of this manuscript.

Reviewer #1 (Remarks to the Author):

The authors responses and revisions to the manuscript with regards to my concerns are satisfactory. I think this paper can be accepted for publication.

Reviewer #2 (Remarks to the Author):

In their second revision, the authors have addressed all of my remaining concerns. I can now recommend the publication of this manuscript.

Authors:

We thank the referees for their insightful suggestions that helped improve our work, and for the positive assessment.